# Vaccines and Autoimmunity—From Side Effects to ASIA Syndrome

**DOI:** 10.3390/medicina59020364

**Published:** 2023-02-14

**Authors:** Isa Seida, Ravend Seida, Abdulrahman Elsalti, Naim Mahroum

**Affiliations:** International School of Medicine, Istanbul Medipol University, 34810 Istanbul, Turkey

**Keywords:** ASIA, adjuvant, hepatitis B vaccine, influenza vaccine, COVID-19

## Abstract

Since vaccines are in fact manufactured chemical compounds such as drugs, the appearance of side effects following their use is not surprising. Similarly, as the main goal of vaccines is to stimulate the immune system bringing out the production of protective antibodies, autoimmune-related side effects as a consequence of increased immune activity do not seem irrational. Fortunately, the rate of such side effects is low; however, the importance of reporting adverse events following vaccinations, understanding the mechanisms behind their appearance, making early diagnosis, and appropriate treatment cannot be overemphasized. In fact, autoimmune-related side effects of vaccines, particularly those based on adjuvants, were reported long before the introduction of the autoimmune/inflammatory syndrome induced by adjuvants (ASIA). Nevertheless, ASIA gathered and united the side effects of vaccines under one title, a step which helped organize the research and call for better immune stimulators than adjuvants. New technologies and methods of making vaccines were clearly noticed during the pandemic of COVID-19 after the introduction of mRNA-based vaccines. In our current paper, we introduce the notion of side effects to vaccines, particularly those of autoimmune nature, the mechanisms of ASIA, and the main vaccines linked with the syndrome including the recent COVID-19 vaccines. The transition from side effects to ASIA is the main idea behind our work.

## 1. Introduction

Vaccines have continued to be a vital preventative measure against infectious diseases for more than 200 years [1,2,3,4]. Not dissimilar to other drugs, vaccines are associated with certain side effects, some of which are short lasting and acute while others are a consequence of complex interactions with the immune system of the host which might ultimately result in the induction of autoimmunity and autoimmune diseases.

Autoimmune diseases are multifactorial disorders manifesting as a consequence of a conglomerate of factors where genetic and environmental ones constitute the core of this correlation [5]. In turn, adjuvants, the chemical compounds utilized in the manufacturing of vaccinations aiming to boost the ability of vaccines to generate an immune response [6], have been addressed for decades as one of the key triggers of autoimmune phenomena. In fact, adjuvants make the potential of autoimmune manifestations secondary to vaccination not surprising. In terms of components, adjuvants range from aluminum-based material often used in vaccines, to silicon and heavy metals such as mercury and iodine gadital, among others [7].

Autoimmune/inflammatory syndrome induced by adjuvants (ASIA), first defined by Shoenfeld et al. in 2011 [8], consists of an immune response that follows the introduction of adjuvants into the body. The syndrome of ASIA consists of a collection of 5 syndromes: macrophagic myofasciitis syndrome, sick building syndrome, Gulf War syndrome, siliconosis, and vaccination-induced autoimmunity [9]. For a condition to be diagnosed as ASIA, two major criteria or one major criteria alongside two minor ones must be fulfilled [10]. Major criteria are represented mainly by clinical aspects of ASIA syndrome, in addition to a positive history of exposure to external stimuli (adjuvants), histological patterns corresponding with typical inflammatory findings, or improvement after removal of the offending subject [11]. In turn, minor criteria refer to the immune related aspects of the syndrome; these include anti-adjuvant antibodies, the development of autoimmune disorders, development of secondary disorders such as IBS, and genetic predisposition via certain HLA-antigens (HLA-DRB1, HLA-DQB1). The criteria are illustrated in Table 1.

We aimed in our current paper to delineate the major lines of autoimmune-related side effects and the transition to a more precise syndromic description of the constellation of symptoms and manifestations induced by adjuvants.

## 2. A Historical Background

The incidence of autoimmune disorders in the context of vaccines had been extensively researched prior to the introduction of the term “ASIA syndrome” mentioned earlier. Along with the Gulf War syndrome and other neurological manifestations, silicone breast implants and tattoos are now included in the category of “autoimmune/inflammatory syndrome generated by adjuvants” (ASIA) [12].

Multiple autoimmune syndromes have been reported in the past in the settings of vaccinations, such as optic neuritis and myelitis following tetanus toxoid vaccine [13], and vasculitis following influenza vaccine in the 1990s [14]. Additionally, many other vaccines have been reported to cause autoimmune disorders, such as cases of immune thrombocytopenic purpura [15] and diabetes mellitus [16] reported after receiving the MMR vaccine. Among the vaccines most associated with autoimmune side effects historically is the hepatitis B vaccine. Hepatitis B vaccine has been associated with erythema nodosum [17], immune thrombocytopenia [18], myasthenia gravis [19], uveitis [20], Reiter’s syndrome [21], arthritis [22], systemic lupus erythematosus (SLE) [23], central nervous system (CNS) demyelination [24], and finally Evan’s syndrome [25].

It is interesting to note that HLA-B27 antigen carriers have been linked to an increased risk of developing autoimmune diseases after vaccination [26], including uveitis, Reiter’s syndrome, and ankylosing spondylitis, denoting and supporting the idea of “the mosaic of autoimmunity” with its complex mechanisms and appearance in genetically susceptible individuals [27]. As per further studies by Shoenfeld et al. in 2009, the immune system recognizes adjuvant molecules through toll-like receptors (TLRs) on leukocytes, inducing an adjuvant-induced immune response. These findings led to the hypothesis that adjuvants, including virosomes for HBV, HPV, and HAV, MF59 in certain viral vaccines, MPL, AS04, AS01B, and AS02A against viral and parasite illnesses, and cholera toxin for cholera, may be the factor that predisposes an individual to autoimmune disorders upon vaccination [28].

Various adjuvants have been described to cause autoimmune diseases. For instance, mineral oil adjuvants have been linked to sclerosing lipogranulomas [29], aluminum adjuvants have been associated with the appearance of multiple sclerosis, chronic fatigue syndrome, and polymyalgia rheumatica [30], and finally, silicone adjuvants were linked to cases of connective tissue diseases [31], scleroderma, SLE, and rheumatoid arthritis [32]. Recently, tattoos have also been suggested to be part of the ASIA syndrome, as they have been connected to cases of sarcoidosis granuloma, which were described following interferon-alpha treatment for a head melanoma in a patient with tattoos [33].

All of the findings described led to the introduction of the term ASIA by Shoenfeld and his team.

## 3. The Pathophysiology of ASIA

When compared to the majority of other autoimmune diseases, ASIA is similar in that several factors underlie its pathogenesis, including environmental and genetic factors. As implied by the name of the syndrome, the main underlying environmental factor is adjuvants which compose the main part in the major criteria of ASIA [12] (Table 1). Adjuvants are agents that boost the activation of the innate immune system resulting in better efficacy of substances such as vaccines [34]. One of the most common adjuvants in use is aluminum and its salts, which have been used in the production of vaccines for tetanus, encephalitis, HPV and many others [35]. Adjuvants, through molecular mimicry [36], act as ligand for TLRs [37], which in turn, once activated, start producing type I INF and proinflammatory cytokines [38]. Moreover, adjuvants lead to the recruitment of dendritic cells via chemotaxis and activation of antigen presenting cells, which in turn results in more portent B-cell and T-cell responses. This ultimately results in a stronger adaptive immune response to antigens [28].

Another environmental factor used since the 1960s, with several reports of autoimmune reactions secondary to its application, is silicone [39]. The latter has been implicated in activation of the adaptive immune cell proliferation and cytokine release that lead to T cell proliferation and polarization which result eventually in promotion of fibrosis [40].

In terms of genetics, although considered a minor criterion for the diagnosis of ASIA (Table 1), it holds an important role in the interplay by providing fertile grounds for ASIA to propagate. When speaking of genetic factors for ASIA and other autoimmune diseases, genes such as HLA-DRB1 and HLA DQB1 [41], alongside PTPN22, take the spotlight of genetic susceptibility to ASIA [42]. Moreover, certain studies suggest epigenetic involvement, because despite the high rates of exposure to environmental factors, susceptibility to ASIA is still rather low [43].

## 4. Major Vaccines Associated with ASIA

### 4.1. Hepatitis B Vaccine

Hepatitis B vaccine is considered to be the vaccine with most autoimmune manifestations among current vaccines. In an analysis of 93 individuals with a mean age of 26.5 years, Zafrir et al. [44] reported symptoms at a mean of 43.2 days following hepatitis B vaccinations. Of the sample, 70% of the participants had neuropsychiatric symptoms, 60% showcased neurological manifestations including photosensitivity 30%, paresthesia 23%, short term memory loss 15%, dizziness 14%, gait disturbance 8.6%, burning sensation 7.5%, paralysis 7.5%, optic neuritis 7.5%, cognitive dysfunction 7.5%, neurogenic bladder and bowel 6.5%, ataxia 6.5%, seizure 5%, nystagmus 4%, vertigo 4%, hyporeflexia 4%, Lhermitte’s phenomenon 3%, hyperesthesia 3%, urinary retention 2%, dysarthria 2%, tinnitus 2%, nuchal rigidity 1%, myoclonic jerks 1%, and tics 1%. Moreover, 32% of the patients also displayed ophthalmological manifestations; among these were eye field visual changes 20.4%, diplopia 6.5%, visual loss 6.4%, uveitis 3%, conjunctivitis 2%, gaze disturbance 2%, and retinopathy 1%. Musculoskeletal signs were demonstrated by 59% of the patients, these symptoms involved arthralgia 36.5%, myalgia 25.8%, joint stiffness 19.3%, back pain 14%, arthritis 10.7%, muscle spasm 7.5%, muscle tone 3%, and muscle wasting 3%. Nausea, vomiting, abdominal pain, weight loss, decreased appetite, diarrhea, and constipation were among the GI symptoms found in 50% of the patients. Skin-related side effects were registered in 30% of patients including rash, malar rash, photosensitivity, and Raynaud’s phenomenon. Finally, 60% of patients displayed general symptoms such fatigue 41.9%, weakness 20.4%, fever 18%, chills 7.5%, and lymph node enlargement 5%.

Certain studies have also reported cases of SLE [45], acute disseminated encephalomyelitis (ADEM) [46], and elevations in anti-phospholipid antibodies [47,48] following the administration of recombinant HBV vaccine.

### 4.2. Influenza Vaccine

Multiple cases have linked influenza vaccinations to autoimmune complications in relation to the endocrine system. A case study demonstrated a 21-year-old Caucasian male patient who presented with adrenal crisis one week following the administration of the influenza vaccine alongside diphtheria, tetanus, and acellular pertussis (DTaP) [49]. In addition, Hsiao et al. reported a case involving a 25-year-old female who developed neck pain and swelling two days following influenza vaccine administration [50]. Fine needle aspiration of the lesion revealed multinuclear giant granulomas in the thyroid gland. Another case of a 36-year-old female demonstrated symptoms of tachycardia, anxiety, and tenderness in her neck one month following the administration of H1N1 influenza vaccine [51]. The patient was later diagnosed with subacute thyroiditis.

Furthermore, numerous studies have linked influenza vaccination to Guillain–Barré syndrome (GBS). For instance, in the 1970s during the influenza vaccination program among military personnel in the US, one case of GBS was described for every 10,000 vaccinated individuals [52]. Additionally, in a meta-analysis of 39 studies published between 1981 and 2014, an increased risk of GBS following the administration of influenza vaccine was found; this increase was particularly prominent in patients receiving H1N1 vaccination [53].

Vasculitis has also been linked to influenza vaccination in several studies. In an evaluation of 45 reports between 1966 to 2016, Watanbe found that 65 patients developed vasculitis in the aftermath of influenza vaccination [54]. Among the cases, 13 were large vessel vasculitis, 42 were small vessel vasculitis, and 5 were single organ vasculitis. Eight cases of giant cell arteritis (GCA) were also reported by Soriano et al. [55]. Moreover, Shoenfeld et al. reported 2 cases of ANCA-associated vasculitis following influenza vaccination [8]. The association was further elaborated in a study that displayed four cases of ANCA-associated vasculitis after influenza vaccinations, these cases included new onset as well as relapses [56].

Influenza vaccinations have also been associated with APS [47], narcolepsy [57], ADEM [46,58], Crohn’s disease [59], and transverse myelitis [60].

### 4.3. Human Papilloma Virus (HPV) Vaccinations

In an analysis of the Vaccine Adverse Event Reporting System (VAERS) database, Geier et al. demonstrated that individuals who were vaccinated with the GARDASIL quadrivalent HPV vaccine were associated with gastroenteritis, SLE, alopecia, CNS conditions, and arthritis among others [61].

HPV vaccination has been associated with menstrual cycle abnormalities and primary ovarian insufficiency (POI). Three unrelated cases of female patients developing menstrual cycle abnormalities in Austria following HPV vaccine administration were reported [62]. Colafrancesco and colleagues described three cases of secondary amenorrhea after HPV vaccination; the patients’ symptoms did not improve with administration of hormone replacement therapy [63].

HPV vaccination has also been associated with SLE [64], and vasculitis [61] including Henoch–Schoenlein purpura [59].

### 4.4. COVID-19 Vaccinations

Since the inception of the SARS-CoV-2 pandemic and its initial warnings [65], vaccines have served as a great factor in curbing the morbidity and mortality of the outbreak. According to VAERS of US Centers for Disease Control and Prevention, more than 550 million doses of SARS-CoV-2 vaccinations have been administered [66]. About 0.0042% major adverse events were reported; these adverse events include severe allergic reactions, thrombotic events, thrombocytopenia, Guillain–Barré syndrome, myocarditis, and death. Moreover, manifestations have also been addressed in autoimmunity-related discussions and seem to increase as vaccination rates rise [67,68,69]. Upon the evaluation of 276 published cases examining side effects of COVID-19 vaccinations, Jara and colleagues [70] found the following to be reported: Guillain–Barré syndrome (151 patients), vaccine-induced thrombotic thrombocytopenia (93 cases), autoimmune liver diseases (8 cases), immune thrombocytopenic purpura (7 cases), IgA nephropathy (5 cases), Graves’ disease (4 cases), systemic lupus erythematosus (3 cases), autoimmune polyarthritis (2 cases), and rheumatoid arthritis (2 cases). The side effects mentioned and reported were mainly related to the mRNA (BNT162b2 or mRNA-1273) and the adenovirus vector-based vaccine ChAdOx1 [68]. Generally speaking, the side effects were rare in comparison to the numbers of people vaccinated, and the newly developed and introduced method, particularly mRNA, has paved the way for safer and more effective vaccines produced in a very short period of time [71,72].

Studies have also reported ADEM [70,73], type 1 diabetes mellitus [74], vasculitis [75,76], and autoimmune thyroid complications [77,78,79,80,81,82,83] of COVID-19 vaccinations.

## 5. Conclusions

Vaccines continue to be absolutely essential in combating infectious diseases, through preventing their transmission and decreasing their associated morbidity and mortality. However, it is important to consider the potential of adverse events related to vaccines particularly those with autoimmune complications especially in genetically predisposed individuals. The latter would doubtlessly aid in prevention, early diagnosis, and treatment. In addition, understanding the risks alongside the mechanisms behind ASIA is crucial for developing vaccines with a safer side effects profile. We view the newer technologies introduced in the development of COVID-19 vaccines, such as the mRNA mechanism, as a good example. This approach is highly welcomed for vaccines as the pandemic contributed to their appearance.

## Figures and Tables

**Table 1 medicina-59-00364-t001:** The diagnostic criteria for ASIA syndrome divided into major and minor criteria.

Major Criteria	Minor Criteria
1- Exposure to external stimuli (infection, vaccine, silicone, adjuvant) before the onset of clinical symptoms.	1- Appearance of antibodies directed against the adjuvant suspected to be involved.
2- The appearance of typical clinical manifestations: a. Myalgia, myositis, or muscle weakness. b. Arthralgia and/or arthritis c. Chronic fatigue, un-refreshing sleep, or sleep disturbances. d. Neurological manifestations (especially associated with demyelination). e. Cognitive impairment, memory loss. f. Fever.	2- Secondary clinical manifestations (irritable bowel syndrome, interstitial cystitis, etc.).
3- Typical histological findings after biopsy of offending organs (such a granulomatous inflammation or lymphocytic infiltration).	3- Evolvement of an autoimmune disease (i.e., MS, SSc).
4- Removal of offending agent results in improvement of symptomatology.	4- Antigens specific for human leukocytes (HLA DRB1, HLA DQB1) linked with the development of ASIA.

SSC = systemic sclerosis; MS = multiple sclerosis.

## Data Availability

PubMed.

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
