# Peer review of "Vaccines and Autoimmunity—From Side Effects to ASIA Syndrome"

_medicina, 2023, doi:10.3390/medicina59020364_

Round 1
Reviewer 1 Report
The present review article entitled “Vaccines and Autoimmunity - from side effects to ASIA syndrome” describes ASIA syndrome based on different diagnostic criteria and disease manifestations. However, a few points are to be taken into consideration to enrich with more factual information in the line of the topic. My comments have been listed below:
Major comments:
1. Page no. 2, Authors should mention the basis of major and minor diagnostic criteria separation with proper references.
2. Authors are requested to include a basic mechanism of molecular mimicry due to ASIA using schematic representation.
3. Is there any studies which show age and sex-based variations in ASIA?
4. Please explain how genetic and environmental factors are affecting ASIA.
5. Please include how to overcome the post-vaccination ASIA complications and future guidelines.
Author Response
Thank you so much for your important comments.
Here are our responses, highlighted in yellow following each point:
- Page no. 2, Authors should mention the basis of major and minor diagnostic criteria separation with proper references.
We re-arranged the text and wrote more on the criteria, and also the table.
- Authors are requested to include a basic mechanism of molecular mimicry due to ASIA using schematic representation.
Molecular mimicry was mentioned in brief with a reference, in the pathophysiology section, with more detailed added on the role of adjuvants in this regard.
- Is there any studies which show age and sex-based variations in ASIA?
Unfortunately, if to exclude breast implants related ASIA, we could not find such studies.
- Please explain how genetic and environmental factors are affecting ASIA.
We added a sentence and a reference on the genetic predisposition.
- Please include how to overcome the post-vaccination ASIA complications and future guidelines.
It is a very important point. In fact, we mentioned the importance of reporting these side effects, working on the understanding of the mechanisms behind them, and viewing ASIA as a uniting entity to better study these side effects and research in this domain. The points were emphasized both throughout the text and in the conclusion.
Reviewer 2 Report
The authors discussed the side effects of vaccine, autoimmune diseases in serious cases. The topic is interesting to readers, however need to include more detailed information, e.g. how different types of Covid-19 vaccines are associated with specific side effects. Also some confusing writing in places.
1. Some major or minor criteria in Table 1 is not clear, e.g.
- what is the typical histological findings?
- Antigens specific for leukocytes, did you mean expressing these genes?
- For major criteria 2, does the patient have to have manifestations of a-f as one major criteria?
2. Improve English throughout the paper, e.g.
- Page 2, “Actually, adjuvants make the potential……not too unlikely.”
- Page 2, “The syndrome of ASIA is made of…”, did you mean the syndrome of ASIA includes a group of conditions:….?
3. When discussing the influenza vaccination side effect, three cases were included, would be good to give more information about the incidence rate.
4. Author included side effects associated with Covid-19 vaccine, need to have more details of what type of vaccines for each side effect, mRNA or others as there should be difference. As the authors summarised in the conclusion that “we view the newer technologies introduced…”, but no detail of newer technologies was discussed in Covid-19 vaccinations.
5. Section numbering is all wrong, every one starts with “1. ….” Or “1.1 ….”.
Author Response
Dear reviewer,
Thank you so much for your valuable comments.
Here are our responses, highlighted in yellow following each point:
- Some major or minor criteria in Table 1 is not clear, e.g.
- what is the typical histological findings?
- Antigens specific for leukocytes, did you mean expressing these genes?
- For major criteria 2, does the patient have to have manifestations of a-f as one major criteria?
We addressed these points in more details in the text and in the table. The edits can be seen in track changes.
- Improve English throughout the paper, e.g.
- Page 2, “Actually, adjuvants make the potential……not too unlikely.”
- Page 2, “The syndrome of ASIA is made of…”, did you mean the syndrome of ASIA includes a group of conditions:….?
English was revised throughout the text.
We meant consists of, we corrected it accordingly.
- When discussing the influenza vaccination side effect, three cases were included, would be good to give more information about the incidence rate.
Unfortunately, most of the literature consists of case reports or series of case reports, or a meta-analysis illustrated in out article. For the best of our knowledge, there is concrete or precise incidence rate.
- Author included side effects associated with Covid-19 vaccine, need to have more details of what type of vaccines for each side effect, mRNA or others as there should be difference. As the authors summarised in the conclusion that “we view the newer technologies introduced…”, but no detail of newer technologies was discussed in Covid-19 vaccinations.
We mentioned the vaccines reported (mRNA and vector-based), and we added several sentences regarding the new methods introduced both in the COVID-19 and conclusion parts. Two more references were added in this regard.
- Section numbering is all wrong, every one starts with “1. ….” Or “1.1 ….”.
We went over it again and arranged it.
Round 2
Reviewer 1 Report
Authors have now modified the manuscript according to the comments as raised.